# Morphopathogenesis of Adult Acquired Cholesteatoma

**DOI:** 10.3390/medicina59020306

**Published:** 2023-02-07

**Authors:** Kristaps Dambergs, Gunta Sumeraga, Māra Pilmane

**Affiliations:** 1Department of Otorhinolaryngology, Riga Stradiņš University, Pilsonu Street 13, LV-1002 Riga, Latvia; 2Children’s Clinical University Hospital, Vienibas Gatve 45, LV-1004 Riga, Latvia; 3Department of Morphology, Institute of Anatomy and Anthropology, Riga Stradiņš University, LV-1007 Riga, Latvia

**Keywords:** cholesteatoma, Ki-67, transcription factors, metalloproteases, vascular endothelial growth factor, cytokines, defensins, Sonic hedgehog, adults

## Abstract

*Background and Objectives*. The aim of this study was to compare the distribution of proliferation markers (Ki-67, NF-κβ), tissue-remodeling factors (MMP-2, MMP-9, TIMP-2, TIMP-4), vascular endothelial growth factor (VEGF), interleukins (IL-1 and IL-10), human beta defensins (HβD-2 and HβD-4) and Sonic hedgehog gene protein in cholesteatoma and control skin. *Methods.* Nineteen patient cholesteatoma tissues and seven control skin materials from cadavers were included in the study and stained immunohistochemically. *Results*. Statistically discernible differences were found between the following: the Ki-67 in the matrix and the Ki-67 in the skin epithelium (*p* = 0.000); the Ki-67 in the perimatrix and the Ki-67 in the connective tissue (*p* = 0.010); the NF-κβ in the cholesteatoma matrix and the NF-κβ in the epithelium (*p* = 0.001); the MMP-9 in the matrix and the MMP-9 in the epithelium (*p* = 0.008); the HβD-2 in the perimatrix and the HβD-2 in the connective tissue (*p* = 0.004); and the Shh in the cholesteatoma’s perimatrix and the Shh in the skin’s connective tissue (*p* = 0.000). *Conclusion*. The elevation of Ki-67 and NF-κβ suggests the induction of cellular proliferation in the cholesteatoma. Intercorrelations between VEGF, NF-κβ and TIMP-2 induce neo-angiogenesis in adult cholesteatoma. The similarity in the expression of IL-1 and IL-10 suggests the dysregulation of the local immune status in cholesteatoma. The overexpression of the Sonic hedgehog gene protein in the cholesteatoma proves the selective local stimulation of perimatrix development.

## 1. Introduction

The worldwide incidence of acquired adult cholesteatoma is from 9 to 12.6 cases per 100,000 adults annually [1]. Histologically, this benign tumor is composed of three parts. The innermost part contains anucleate epithelial cells, which form a cystic layer. The second part is the hyperproliferative squamous epithelial layer—the matrix. The outer part, the perimatrix, is a granulation tissue rich in different inflammatory cells [2]. Although cholesteatoma is benign, it acts destructively towards the surrounding tissue in the temporal bone [3]. However, the complex etiopathogenesis of cholesteatoma is uncertain.

Since cholesteatoma is constantly proliferating, the Ki-67 is one of the most frequently used markers to detect proliferation in cholesteatoma tissue [4]. Recent studies suggest that Ki-67 could be a prognostic factor for cholesteatoma’s destructiveness in the middle ear. However, this marker cannot predict the recurrence of cholesteatoma [5]. Another cell factor that causes keratinocyte proliferation in cholesteatoma is the nuclear factor kappa beta (NF-κβ) [6]. It was shown that NF-κβ prevents epithelial cells in the cholesteatoma matrix from entering apoptosis and, therefore, that the growth and expansion of the cholesteatoma is supported by NF-κβ via this mechanism as well [7]. Additionally, Hamajima et al. [7] demonstrated that Ki-67 and NF-κβ act together to induce the proliferation process in the matrix.

Further, because the local osteolytic process in the middle ear is a characteristic pattern for cholesteatoma patients, we chose to evaluate several remodeling factors, such as matrix metalloproteinase 2 (MMP-2), matrix metalloproteinase 9 (MMP-9) and the tissue inhibitors of the metalloproteinases 2 and 4 (TIMP-2; TIMP-4). The remodeling factors MMP-2 and MMP-9 are gelatinases and are responsible for angiogenesis and preparation for remodulation of the bone. In an inflammatory environment, such as chronic otitis media with cholesteatoma, MMP-2 and MMP-9 cause pathologic bone degradation [8,9]. On the other hand, TIMPs are regulators of MMPs, and they form stoichiometric complexes [10], which are important for sustaining homeostasis in organisms. The interruption of the balance between MMPs and TIMPs was proven to start or continue the progression of different diseases, including cholesteatoma [11,12]. Both TIMP-2 and TIMP-4 inhibit MMP-2 and MMP-9 [11].

Increased neo-angiogenesis in the perimatrix is important for cholesteatoma expansion [13]. Vascular endothelial growth factor (VEGF) is a very active angiogenic agent whose expression is more active in hypoxia [14]. Hypoxia is crucial for the pathogenesis of chronic middle-ear infection [15]. The overexpression of VEGF in cholesteatoma compared to control skin has been linked to its aggressiveness [16].

Acquired middle-ear cholesteatoma is often accompanied by the presence of different bacteria [17]. Therefore, we chose to evaluate interleukin-1 (IL-1), interleukin-10 (IL-10) and human beta defensins 2 and 4 (HβD-2, HβD-4). Different studies proved that IL-1 is associated with cholesteatoma growth (via the induction of keratinocyte proliferation) and the bone-erosion process [18,19,20]. By contrast, interleukin 10 is the most active anti-inflammatory cytokine, and its impaired expression can exacerbate the inflammatory response and lead to tissue damage [21]. The disproportion between IL-1 and IL-10 in cholesteatoma could be one of many causes of its aggressiveness [22]. Furthermore, human beta defensins 2 and 4 reduce the inflammation caused selectively by Ps. Aeruginosa [23,24]. It was proven that HβD-2, from all the defensins plays a major role in the pathophysiology of middle-ear diseases [25]. However, additional information on how HβD-2 acts in cholesteatoma tissue is needed. There are limited data about the importance of HβD-4 in cholesteatoma, but our previous study showed that HβD-2 is more heavily expressed in pediatric cholesteatoma tissue than HβD-4 [26].

Lastly, we chose to study the Sonic hedgehog (Shh) gene protein distribution in cholesteatoma and control skin tissue. Sonic hedgehog encodes the pharyngeal endoderm and directly controls the early development of the middle ear [27]. Chiang et al. proved that the loss of Shh causes external- and middle -ear pathologies [28]. However, the exact role of Shh in cholesteatoma tissue is not yet clear, although our previous study showed the upregulation of Shh in cholesteatoma compared to skin tissue [26].

Thus, given the complexity of the pathogenesis of cholesteatoma, the aim of this study was to compare the distribution of proliferation markers (Ki-67, NF-κβ), tissue-remodeling factors (MMP-2, MMP-9, TIMP-2, TIMP-4), vascular endothelial growth factor (VEGF), pro-and anti-inflammatory cytokines (IL-1, IL-10), defensins (HβD-2, HβD-4) and Shh gene protein in cholesteatoma and control skin tissue.

## 2. Materials and Methods

### 2.1. Tissue Samples

This study was conducted between November 2019 and December 2021. The study was approved by the Ethical Committee of the Riga Stradiņš University (5 September 2019; no. 6-2/7/4) and performed according to the 2013 Declaration of Helsinki. All the patients gave their informed consent to participate in the study. The nature of the study was fully explained to the patients.

Tissue samples were collected during cholesteatoma surgery in P. Stradiņš Clinical University Hospital by one surgeon, and the immunohistochemical staining and analysis of the tissues were performed at the Department of Morphology of the Riga Stradiņš University, Latvia.

In the patient group, 24 patients participated in this study, 11 male and 13 female (ages varied from 19 to 75 years, mean age 36, 37 years). Inclusion criterion was acquired adult cholesteatoma. Five patients were excluded from the study. The exclusion reasons were incomplete cholesteatoma material (cholesteatoma without matrix and/or perimatrix), which was invalid for immunohistochemical analysis.

In the control group, ten deep external meatal skin-tissue samples were obtained from 10 different cadavers (ages ranging from 35 to 70 years) in a collection from the Institute of Anatomy and Anthropology. The use of the cadaver material was approved by the Ethical Committee of the Riga Stradiņš University (29 October 2022; 2-PĒK-4/475/2022). Inclusion criterion was adults with no chronic ear or skin diseases. Three control-group skin samples were excluded because of insufficient skin material, which was invalid for immunohistochemical analysis.

### 2.2. Immunohistochemical Analysis

Tissues were fixed in a mixture of 2% formaldehyde and 0.2% picric acid in 0.1 M of phosphate buffer (pH 7.2). Subsequently, they were rinsed in Tyrode buffer (content: NaCl, KCl, CaCl_2__2H_2_O, MgCl_2__6H_2_O, NaHCO_3_, NaH_2_PO_4__H_2_O, glucose) containing 10% saccharose for 12 h and then embedded into the paraffin.

Thin sections (3 µm) were cut. Xylene (BC-5L, Biognost, Zagreb, Croatia) was used to clear off paraffin and alcohol 96° to dehydrate tissue sections. The slides were prepared for histological routine staining and immunohistochemical staining using the HiDef Detection™ HRP Polymer System (954D-30, Cell Marque, Rocklin, CA, USA) to identify the following markers in tissue samples: Ki-67 (obtained from rabbit, 1325506A, 1:100, Cell Marque, Rocklin, CA, USA); NF-κβ (NFkB-105; obtained from rabbit, 1:100 dilution, Abcam, Cambridge, U.K.); MMP2 (mouse, sc-53630, 1:100, Santa Cruz Biotechnology, Inc., Dallas, TX, USA); matrix metalloproteinase-9 (MMP-9) (sc-10737, rabbit, working dilution 1:100, Santa Cruz Biotechnology, Inc., Santa Cruz, Dallas, TX, USA); TIMP2 (mouse, sc-21735, 1:200, Santa Cruz Biotechnology, Inc., Dallas, TX, USA); tissue inhibitor of metalloproteinase-4 (TIMP-4) (at 1:100 sc-30076, rabbit, working dilution 1:100, Santa Cruz Biotechnology, Inc., Santa Cruz, Dallas, TX, USA); vascular endothelial growth factor (VEGF) (orb191500, rabbit, polyclonal, working dilution 1:100, Biorbyt Ltd., Cambridge, UK); IL-1 (orb308737, rabbit, working dilution 1:100, Biorbyt Limited, Cambridge, UK); IL-10 (orb100193, rabbit, working dilution 1:600, Biorbyt Limited, Cambridge, UK); HβD-2 (sc-20798, working dilution 1:100, Santa Cruz Biotechnology, Inc., Dallas, TX, USA); HβD-4 (ab14419, mouse, working dilution 1:200, Abcam, San Francisco, CA, USA); Shh (LS-C49806, mouse, 1:100, LifeSpan BioSciences, Inc., Seattle, WA, USA).

Next step included rinsing of tissue samples in wash buffer (TRIS; T0083, Diapath S.p.A., Martinengo, Italy) two times for 5 min, followed by placing them in a microwave oven for up to 20 min in boiling EDTA buffer (T0103, Diapath S.p.A., Martinengo, Italy) and then cooling them down to 65 °C (~20 min). The specimen was placed in a TRIS wash buffer and blocking with 3% peroxidase block (925B-02, Cell Marque, Rocklin, CA, USA) was performed for 10 min. All antibodies used in research were diluted with Antibody Diluent (938B-05, Cell Marque, Rocklin, CA, USA).

The HiDef DetectionTM HRP polymer system (954D-30, Cell Marque, Rocklin, CA, USA) was used for the antibodies of mouse or rabbit origin. Slides were rinsed 5 times (5 min each) with TRIS buffer solution. Next, HiDef DetectionTM reaction amplifier (954D-31, Cell Marque, Rocklin, CA, USA) was applied for 10 min at room temperature. After this processing, the preparations were rinsed five times (for five minutes each time) in distilled water. After rinsing, HRP chromogen (used with DAB Buffer) (957D-30, Cell Marque, Rocklin, CA, USA) was used for 3–5 min. Chromogen was made fresh for each application. Subsequently, slides were rinsed 5 times with TRIS buffer solution. Next, slides were placed in a slide basket and immersed in filtered hematoxylin for 30–60 s. After staining with hematoxyline, the micro-preparations were rinsed in distilled water five times and dehydrated in alcohols (at 95% and 100% for 3 min), after which they were immersed in 3 containers with Xylene (5 min each), dried and covered with glue Pertex^®^ (00801-EX, HistoLab, Västra Frölunda, Sweden) glue. Positive controls in accordance with the companies guidelines and negative controls (Appendix A) with exclusion of primary antibody were developed.

The slides were analyzed by light microscopy by two independent morphologists using semi-quantitative method. The results were evaluated by grading the appearance of positively stained cells in the visual field; multiple sections for each sample were scored. Structures in the visual field were labeled as follows: 0 = no positive structures, 0/+ = occasional positive structures, + = few positive structures, +/++ = small-to-moderate number of positive structures, ++ = moderate number of positive structures, ++/+++ = moderate-to-numerous positive structures, +++ = numerous positive structures, +++/++++ = numerous-to-abundant structures, ++++ = an abundance of positive structures in the visual field [29] (Appendix A).

For a visual illustration, a Leica DC 300F digital camera and image-processing-and-analysis software, Image-Pro Plus (Media Cybernetics, Inc., Rockville, MD, USA) were used.

### 2.3. Statistical Analysis

The data processing was performed with SPSS software, version 27.0 (IBM Company, Chicago, IL, USA). In SPSS analysis, no positive structures (0) in visual field were labeled as “0”(0 = 0); 00/+ = 0.33, 0/+ = 0.5; 0/++ = 0.66; + = 1; +/++ = 1.5; ++ = 2; ++/+++ = 2.5; +++ = 3; +++/++++ = 3.5; ++++ = 4. Since the data were ordinal, we used nonparametric tests, Spearman’s rank correlation and Mann–Whitney U test.

Spearman’s rank correlation coefficient was used to determine correlations between factors, where r = 0–0.2 was assumed as a very weak correlation, r = 0.2–0.4 a weak correlation, r = 0.4–0.6 a moderate correlation, r = 0.6–0.8 a strong correlation and r = 0.8–1.0 a very strong correlation. The Mann–Whitney U test was used to analyze the control group versus the patient data. The level of significance for tests was chosen as 5% (*p*-value < 0.05).

## 3. Results

### 3.1. Description of the Tissue

In the routine histological examination with hematoxylin and eosin (H-E), all three parts of the cholesteatoma were visualized. The outer part, the perimatrix, was composed of different inflammatory cells, including lymphocytes, leukocytes, macrophages, as well as fibrocytes, collagen fibers and blood vessels. The middle part was hyperproliferative epithelium, known as the matrix. The inner part, the cystic layer, was an anucleate keratin mass (Figure 1a). The control tissue was deep-external-ear-canal skin that presented an intact stratified squamous epithelium and connective tissue without inflammation (Figure 1b).

### 3.2. Description of Immunohistochemical (IHC) Findings

The TIMP-4, MMP-2 Shh, IL-10 and NF-κβ were the most widely distributed tissue factors in the patient group. The least distributed were MMP-9, HβD-4, Ki-67 and TIMP-2. The most variable tissue factors were IL-1 and IL-10, where marker-positive cells varied from none (0) to numerous (+++) in the matrix and perimatrix (Table 1).

In the control group, the most widely distributed tissue factors were VEGF, IL-10, TIMP-2 and Shh and the least were Ki-67, IL-1, NF-κβ, HβD-2 and HβD-4. The most variable tissue factors were Shh, where Shh containing epithelial cells varied from none (0) to numerous-to-abundant (+++/++++), MMP-2 from none (0) to numerous (+++) in the epithelial layer and TIMP-2 in the epithelium, where TIMP-2 marked a variance from a few (0/+) to moderate-to-numerous (++/+++) factor-positive cells (Table 1).

#### 3.2.1. IHC Findings of Proliferation Markers

The Ki-67-positive cells varied from no (0) positive cells to few-to-moderate (+/++) positive cells in cholesteatoma, but there were mostly no Ki-67 positive cells in the control group.

The appearance of the NF-κβ-containing cells was graded from no (0) to numerous (+++) positive cells in the cholesteatoma and from no (0) to moderate (++; Figure 2a–d).

#### 3.2.2. IHC Findings on the Angiogenetic Factor

The vascular endothelial growth factor in the cholesteatoma presented a variance from no (0) to numerous (+++) positive cells. A similar distribution was seen in the control group (Figure 3a,b).

#### 3.2.3. IHC Findings on the Tissue-Remodeling Factors

A range from no (0) to numerous-to-abundant (+++/++++) of MMP-2-containing cells was detected in the patient group, but in the control group, a range from no (0) to moderate-to-numerous (++/+++) MMP-2-containing cells was found.

The distribution and appearance of the MMP-9-positive cells ranged from none (0) to moderate (++) in the cholesteatoma and from occasional (0/+) to few-to-moderate (+/++) in the skin epithelium (Figure 4a–d).

The numbers of TIMP-2 positive cells in the patient group varied from none (0) to numerous (+++) and, in control group, from none (0) to moderate-to-numerous (++/+++).

The TIMP-4-containing cells in the cholesteatoma displayed a variance from none (0) to numerous-to-abundant (+++/++++) and, in the skin epithelium, from a few (+) to numerous (+++) TIMP-4-containing cells (Figure 5a–d).

#### 3.2.4. IHC Findings of Pro- and Anti-Inflammatory Cytokines and Defensins

The IL-1- and IL-10-positive cells in the patient group varied from none (0) to numerous (+++). In the control group, the IL-1-positive cells ranged from none (0) to few-to-moderate (+/++), and the IL-10-positive cells ranged from occasional (0/+) to moderate-to-numerous (++/+++; Figure 6a,d).

The distribution of the tissue-defensin-HβD-2-containing cells was graded from none (0) to moderate-to-numerous (++/+++) and the HβD-4-positive cells displayed a variance from no (0) to numerous (+++) positive cells. However, in the control group, the number of HβD-2- and HβD-4-positive cells ranged from none (0) to moderate (++) (Figure 7a–d).

#### 3.2.5. IHC Findings of Shh Gene Protein

In the patient group, the Shh findings demonstrated a range from occasional (0/+) to abundant (++++) positive cells, but in the control group, the range was from none (0) to numerous-to-abundant (+++/++++; Figure 8a,b).

### 3.3. Statistical Analysis

To determine the difference between the groups, we used a Mann–Whitney U test.

There were statistically significant differences between the following: the Ki-67 in the matrix and the Ki-67 in the skin epithelium (*p* = 0.000); the Ki-67 in the perimatrix and the Ki-67 in the connective tissue (*p* = 0.010); the NF-κβ in the matrix and the NF-κβ in the control epithelium (*p* = 0.001); the MMP-9 in the matrix and the MMP-9 in the epithelium (*p* = 0.008); the HβD-2 in the perimatrix and the HβD-2 in the control connective tissue (*p* = 0.004); and the Shh in the perimatrix and the Shh in the control connective tissue (*p* = 0.000) (Table 2).

In the patient group, there were moderate correlations between the following: the MMP-2 and the TIMP-2 in the matrix (r = 0.482, *p* = 0.037); the MMP-9 and the TIMP-2 in the perimatrix (r = 0.466, *p* = 0.044); the MMP-9 in the matrix and the TIMP-4 in the perimatrix (r = 0.497, *p* = 0.030); the MMP-9 and the TIMP-4 in the perimatrix (r = 0.558, *p* = 0.013); theTIMP-2 and the TIMP-4 in the perimatrix (r = 0.523, *p* = 0.021); the TIMP-2 and the Shh in the matrix (r = 0.543, *p* = 0.016); the TIMP-2 in the perimatrix and the IL-1 in the matrix (r = 0.587, *p* = 0.008); the TIMP-2 in the perimatrix and the IL-1 in the perimatrix (r = 0.513, *p* = 0.025); the IL-1 in the perimatrix and the IL-1 in the matrix (r = 0.583, *p* = 0.009); the TIMP-4 and the IL-10 in the perimatrix (r = 0.580, *p* = 0.009); the TIMP-2 and the NF-κβ in the matrix (r = 0.592, *p* = 0.008); the TIMP-2 in the perimatrix and the NF-κβ in the matrix (r = 0.507, *p* = 0.027); the TIMP-4 and the NF-κβ in the matrix (r = 0.518, *p* = 0.023); the MMP-9 in the matrix and the NF-κβ in the perimatrix (r = 0.499, *p* = 0.030); the MMP-9 and the NF-κβ in the perimatrix (r = 0.563, *p* = 0.012); the TIMP-4 and the Ki-67 in the perimatrix (r = 0.579, *p* = 0.009); the IL-1 and the Ki-67 in the perimatrix (r = 0.585, *p* = 0.009); the NF-κβ and the Ki-67 in the perimatrix (r = 0.538, *p* = 0.017); the IL-10 and the VEGF in the matrix (r = 0.588, *p* = 0.008); the NF-κβ in the perimatrix and the VEGF in the matrix (r = 0.512, *p* = 0.025); the TIMP-2 in the matrix and the VEGF in the perimatrix (r = 0.581, *p* = 0.009); the TIMP-2 and the VEGF in the perimatrix (r = 0.549, *p* = 0.015); the TIMP-4 in the matrix and the VEGF in the perimatrix (r = 0.475, *p* = 0.040); the IL-1 in the perimatrix and the HβD-2 in the matrix (r = 0.463, *p* = 0.046); the TIMP-2 and the HβD-2 in the perimatrix (r = 0.491, *p* = 0.033); the Shh and the HβD-2 in the perimatrix (r = 0.456, *p* = 0.050); the IL-10 in the matrix and the HβD-2 in the perimatrix (r = 0.583, *p* = 0.009); the Ki-67 and the HβD-2 in the perimatrix (r = 0.520, *p* = 0.022); the HβD-2 in the matrix and the HβD-2 in the perimatrix (r = 0.577, *p* = 0.010); the MMP-2 and the HβD-4 in the matrix (r = 0.538, *p* = 0.018); the Shh and the HβD-4 in the matrix (r = 0.483, *p* = 0.036), the NF-κβ and the HβD-4 in the matrix (r = 0.520, *p* = 0.022); the Shh in the matrix and the HβD-4 in the perimatrix (r = 0.525, *p* = 0.021); and the Shh and HβD-4 in the perimatrix (r = 0.474, *p* = 0.040; Table 3).

Strong correlations were found between the following: the MMP-2 in the matrix and the MMP-2 in the perimatrix (r = 0.626, *p* = 0.004); the MMP-9 in the matrix and the MMP-9 in the perimatrix (r = 0.635, *p* = 0.004); the TIMP-2 in the matrix and the TIMP-2 in the perimatrix (r = 0.697, *p* = 0.001); the MMP-2 and the Shh in the matrix (r = 0.702, *p* = 0.001); theTIMP-4 and the IL-1 in the perimatrix (r = 0.631, *p* = 0.004); the TIMP-2 in the perimatrix and the IL-10 in the matrix (r = 0.643, *p* = 0.003); the IL-1 and the IL-10 in the perimatrix (r = 0.600, *p* = 0.007); the TIMP-2 and the NF-κβ in the perimatrix (r = 0.624, *p* = 0.004); the TIMP-2 and the IL-10 in the perimatrix (r = 0.734, *p* = 0.000); the IL-1 in the matrix and the IL-10 in the perimatrix (r = 0.777, *p* = 0.000); the IL-1 in the matrix and the NF-κβ in the perimatrix (r = 0.627, *p* = 0.004); the IL-1 and the NF-κβ in the perimatrix (r = 0.674, *p* = 0.002); the IL-10 in the matrix and the NF-κβ in the perimatrix (r = 0.690, *p* = 0.001); the IL-10 and the NF-κβ in the perimatrix (r = 0.779, *p* = 0.000); the Shh in the perimatrix and the Ki-67 in the matrix (r = 0.683, *p* = 0.001); the IL-10 in the perimatrix and the VEGF in the matrix (r = 0.674, *p* = 0.002); the TIMP-2 in the perimatrix and the HβD-2 in the matrix (r = 0.616, *p* = 0.005); the IL-10 in the perimatrix and the HβD-2 in the matrix (r = 0.694, *p* = 0.001); the NF-κβ in the perimatrix and the HβD-2 in the matrix (r = 0.654, *p* = 0.002); the IL-10 and the HβD-2 in the perimatrix (r = 0.663, *p* = 0.002); the NF-κβ and the HβD-2 in the perimatrix (r = 0.692, *p* = 0.001); the Ki-67 in the matrix and the HβD-2 in the perimatrix (r = 0.618, *p* = 0.005); the HβD-2 and the HβD-4 in the perimatrix (r = 0.650, *p* = 0.003); and the HβD-4 in the matrix and the HβD-4 in the perimatrix (r = 0.640, *p* = 0.003; Table 3).

**Very strong correlations** were found between the following: the IL-1 and the IL-10 in the matrix (r = 0.820, *p* = 0.000); the IL-10 in the matrix and the IL-10 in the perimatrix (r = 0.839, *p* = 0.000); the TIMP-4 and the NF-κβ in the perimatrix (r = 0.804, *p* = 0.000); the IL-1 and the HβD-2 in the matrix (r = 0.822, *p* = 0.000); and the IL-10 and the HβD-2 in the matrix (r = 0.841, *p* = 0.000; Table 3). Additional Statistics can be found in the Appendix A.

## 4. Discussion

To prove the hyperproliferative activity of cholesteatoma cells compared to the control skin cells, we used Ki-67 and NF-κβ. Several authors presented the upregulation of Ki-67 in cholesteatoma [30,31,32]. Our results were similar and showed a statistically significant overexpression of Ki-67 in the matrix (*p* = 0.000) and perimatrix (*p* = 0.010) compared to the epithelium and connective tissue of the skin. However, controversies about this matter exist; for example, Kuczkowski et al. [33] presented a study in which the difference between the Ki-67 in cholesteatoma and in a control group was not statistically significant. Our results are substantiated by those of Heenen et al. [34], who showed limited Ki-67 activity in an unchanged epidermis and proved that these cells were in a non-proliferative state. In addition there is controversy as to whether Ki-67 overexpression is associated with the level of bone resorption. Hamed et al. [35], Juhász et al. [36] and Mallet et al. [37] concluded that cholesteatomas that cause more bone erosion have a higher expression of Ki-67 compared to those that cause less destruction. By contrast, Aslier et al. [2] did not observe a correlation between bone erosion and the expression of Ki-67.

Further, we observed a statistically discernible difference between the number of NF-κβ containing cells in the cholesteatoma matrix (*p* = 0.001) compared to the skin epithelium, but not in perimatrix (*p* = 0.055) compared to the connective tissues of the skin. However, the *p*-value is very close to being statistically significant, and it might be a tendency. Our results are supported by Byun et al. [6], who found increased levels of NF-κβ compared to retro-auricular skin. Our results show a moderate correlation between Ki-67 and NF-κβ (r = 0.538, *p* = 0.017). The explanation for these results is that Ki-67 and NF-κβ act through the same pathway (the inhibitor of the DNA binding protein 1 (Id1)→NF-κB→cyclin D1→Ki-67) to induce keratinocyte proliferation [7]. Therefore, we conclude that Ki-67 and NF-κβ can be used as proliferation markers in cholesteatoma to prove the existence of a pathologic proliferation stage in cholesteatoma cells compared to control skin.

Another characteristic pattern in acquired middle-ear cholesteatoma is a local osteolytic process in the temporal bone [38]. The remodeling factors MMP-2 and MMP-9 are associated with bone remodulation in the middle ear for patients with cholesteatoma [8,9]. In contrast to Morales et al. [39] and Olszewska et al. [40], who presented the overexpression of MMP-9 and MMP-2 in cholesteatoma in opposition to retro-auricular skin, we presented no statistically discernible differences between MMP-2 in the patient and the control groups in the soft tissues. These findings are similar to those of Banerjee [41], who did not find differences in the expression of MMP-2 in cholesteatoma and deep meatal skin. Additionally, our results presented statistically significant reduced relative numbers of MMP-9 positive cells in the matrix (*p* = 0.008) compared to the skin epithelium. Limited data exist on the role of decreased levels of MMP-9 in cholesteatoma or any other pathology in humans. However, Pozzi et al. [42] concluded that reduced levels of MMP-9 are associated with increased tumor angiogenesis. We can speculate as to the specific decreased-expression pattern of MMPs in cholesteatoma soft tissue, but the lack of studies on this topic prevents us from expanding the role of decreased MMP-9 in cholesteatoma tissue. Further and more specific studies are needed.

Furthermore, our results did not show differences between the relative numbers of TIMP-2 and TIMP-4 between both study groups. Nevertheless, the difference between the TIMP-2 in the matrix and that in the skin epithelium was nearly statistically discernible (*p* = 0.055) and was decreased in the cholesteatoma tissue, in contrast to the skin. Kaya et al. [12] also proved the downregulation of TIMP-2 in cholesteatoma compared to healthy tissue. We suggest that decreases in the activity of TIMP-2 might affect MMPs and cause an imbalance between MMPs and TIMPs. This imbalance can cause proteolysis in the extracellular matrix, which causes bone remodulation in cholesteatoma patients [8].

Neo-angiogenesis has a key role in cholesteatoma expansion [13]. However, our results did not present statistically discernible differences in the expression of VEGF between the patient and the control group. We suggest that increases in the expression of VEGF do not occur in developed blood vessels. It is known that TIMP-2 not only inhibits MMP-2 and MMP-9 but also inhibits neo-angiogenesis and VEGF directly as a separate function from inhibiting MMPs [43,44]. Our results showed a moderate positive correlation between the VEGF and the TIMP-2 (r = 0.581, *p* = 0.009) and decreased levels of TIMP-2 compared to the control skin. Therefore, we suggest that TIMP-2 intercorrelates with VEGF in cholesteatoma tissue and that reduced relative numbers of TIMP-2 mean reduced anti-angiogenetic properties in cholesteatoma tissue, which results in increased neo-angiogenesis. In addition, we found a positive correlation between the VEGF and the NF-κβ (r = 0.512, *p* = 0.025). Several authors proved that NF-κβ can regulate VEGF in cholesteatoma tissue [7,16]. These findings mark the complexity of angiogenesis in cholesteatoma.

We chose HβD-2, HβD-4, IL-1 and IL-10 to evaluate the inflammatory process in cholesteatoma. The results for the IL-1 and the IL-10 were not statistically discernible between the patient and control groups. These results are supported by those of Yetiser et al. [45] and Kuczkowski et al. [18], who also reported no difference between their patient and control groups for IL-1 and IL-10, respectively. Importantly, we found a very strong positive correlation between the IL-1 and IL-10 in the matrix and perimatrix (r = 0.820, *p* = 0.000) and, by contrast, opposition, a very strong negative correlation between the IL-1 and the IL-10 in the control group (r = −0.829, *p* = 0.021). These results might suggest that there is dysregulation between pro- and anti-inflammatory cytokines (IL-1 and IL-10) in cholesteatoma, which causes local inflammation in the middle ear. Moreover, our results showed a statistically significant overexpression of HβD-2 (*p* = 0.004) in the patient group, in contrast to the controls. However, the differences between both groups for HβD-4 were not statistically discernible. Similarly, the upregulation of HβD-2 in cholesteatoma versus the skin was shown by Song et al. [46] and Park et al. [24]. Furthermore, HβD-2 was more actively expressed in cholesteatoma tissue than in HβD-4. Interestingly, Song et al. [46] showed that HβD-2 is more expressed in cholesteatoma tissue than in HβD-3. Therefore, we might speculation that HβD-2 is most active in human beta defensins against bacterial infection, but more studies are needed to confirm this. Additionally, we found very strong positive correlations between HβD-2 and IL-1 (r = 0.822, *p* = 0.000), as well as strong positive correlations between HβD-2 and NF-κβ (r = 0.692, *p* = 0.001) and NF-κβ and IL-1 (r = 0.674, *p* = 0.002). These findings can be explained by the fact that IL-1 stimulates the production of HβD-2, and that this process is activated by NF-κβ [47,48]. Furthermore, Kanda et al. [49] presented positive correlations between IL-10 and HβD-2 and concluded that HβD-2 increases the production of IL-10 in T cells. We showed a similar correlation between HβD-2 and IL-10 (r = 0.663, *p* = 0.002) in the cholesteatoma perimatrix, where T cells predominate [50]. We did not find similar correlations between HβD-4 and IL-1, or between IL-10 and NF-κβ, as was the case with HβD-2. Therefore, we conclude that HβD-2 is a more potent antibacterial peptide in cholesteatoma than HβD-4.

Finally, our results demonstrated the statistically discernible overexpression of the Shh gene protein in the perimatrix (*p* = 0.000), in contrast to the control group. There are limited data available on Shh’s role in cholesteatoma tissue. Our previous research, which compared children’s cholesteatoma and deep meatal skin controls, showed similar findings to our current study [26]. However, it is known that Shh is responsible for developing the first pharyngeal arch, the pharyngeal endoderm, as well as regulating Fgf8 in the ectoderm from which the middle and external ear develop [51,52]. Furthermore, it has been proven that the loss of the Shh gene causes middle- and outer-ear pathologies [28]. We suggest that Shh is involved in the postnatal stimulation of endodermal/mesodermal tissue.

However, we understand the limitations of our study, namely the relatively small control group and the fact that tissue material were taken from cadavers. However, the ethical considerations mandated the use of this control group. Furthermore, standardized laboratory measurements (e.g., ELISA) could be useful in the evaluation of IHC-stained samples.

## 5. Conclusions

The elevation of Ki-67 and NF-κβ in cholesteatoma tissue suggests the induction of cellular proliferation in this tumor, with the significant involvement of NF-κβ in this process.

The decrease in degradation enzymes and the similarity in the expression of TIMPs might cause pathologic remodulation in cholesteatoma tissue. Intercorrelations between TIMP-2, NF-κβ and VEGF induce neo-angiogenesis in adult cholesteatoma.

The similarity in the expression of pro- and anti-inflammatory cytokines in cholesteatoma suggests the possible stagnation (dysregulation) of the local immune status, which was demonstrated by the strong stimulation of HβD-2 and the intercorrelation between IL-1, NF-κβ and HβD-2. The stimulation of the antibacterial activity of HβD-2 is also not excluded.

The overexpression of the Shh gene protein in cholesteatoma indicates the selective local stimulation of perimatix development, which probably connected to the influence of the endodermal gene protein.

Our study shed light on the complexity of cholesteatoma pathogenesis, as we presented how the aforementioned cell factors intercorrelate. For future studies, our aim is to compare pediatric- and adult-cholesteatoma material to determine the differences between the cell factors in these groups, as well as to increase the number of subjects in the groups.

## Figures and Tables

**Figure 1 medicina-59-00306-f001:**
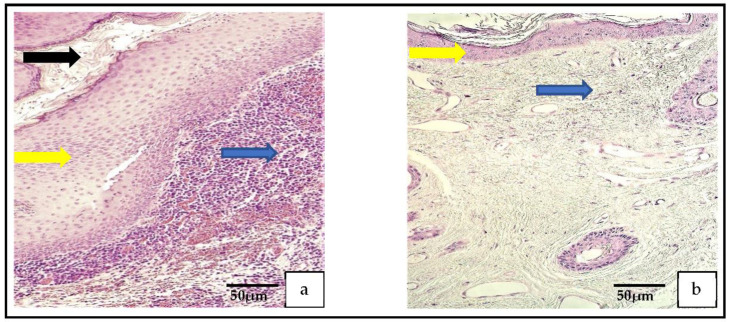
(**a**) Histological micrograph of cholesteatoma. The cholesteatoma matrix (yellow arrow) is the hyperproliferative stratified squamous epithelium. Perimatrix (blue arrow) consists mostly of inflammatory cells and the cystic layer (black arrow) consists of desquamated, anucleate keratin mass. Hematoxylin and eosin. (**b**) Histological micrograph of external ear skin demonstrates unchanged epithelium (yellow arrow) and subepithelial connective tissue (blue arrow) with hair follicles. Hematoxylin and eosin.

**Figure 2 medicina-59-00306-f002:**
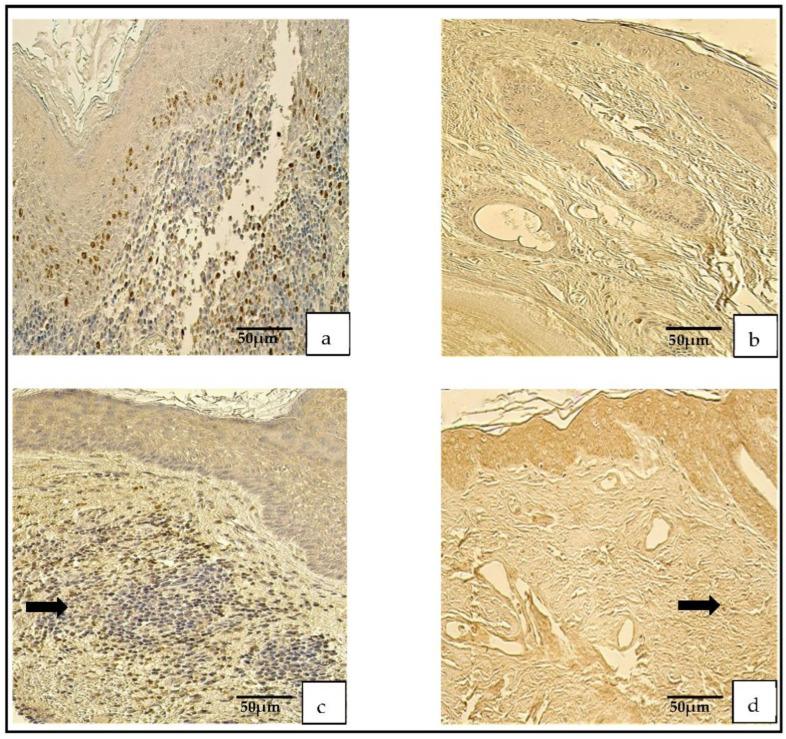
Immunohistochemical micrographs. (**a**) **Cholesteatoma**. A few Ki-67-positive cells in the matrix and the perimatrix, Ki-67 IHC. (**b**) **Skin.** Absence of Ki-67-positive cells in the epithelium and the connective tissue (arrows), Ki-67 IHC. (**c**) **Cholesteatoma**. Moderate-to-numerous NF-κβ-positive cells in the matrix and a few in the perimatrix (arrow), NF-κβ IHC. (**d**) **Skin.** Moderate NF-κβ-positive cells in the epithelium and a few (arrow) in the connective tissue, NF-κβ IHC.

**Figure 3 medicina-59-00306-f003:**
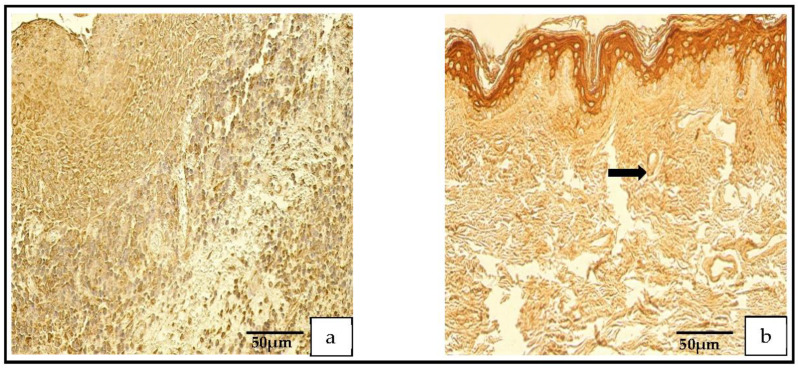
(**a**,**b**) Immunohistochemical micrographs. (**a**) **Cholesteatoma.** Numerous VEGF-positive cells in the matrix and moderate VEGF-positive endothelial cells in the perimatrix, VEGF IHC. (**b**) **Skin.** Numerous VEGF-positive cells in the epithelium and a few VEGF-positive endothelium cells of connective tissue (arrow), VEGF IHC.

**Figure 4 medicina-59-00306-f004:**
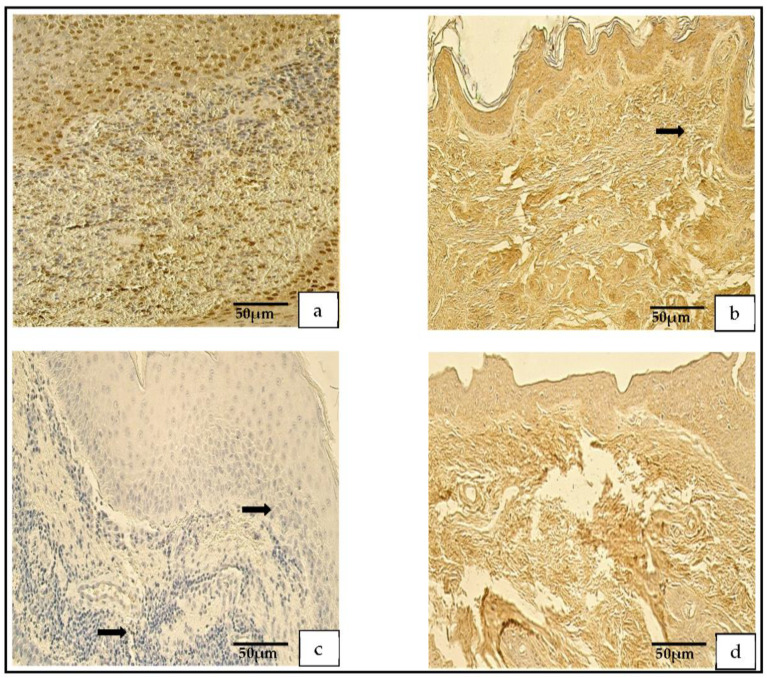
(**a**–**d**) Immunohistochemical micrographs. (**a**) **Cholesteatoma.** Numerous-to-abundant MMP–2-positive cells in the matrix and numerous in the perimatrix, MMP–2 IHC. (**b**) **Skin**. Numerous MMP–2-positive cells in the epithelium, a few (arrow) in the connective tissue, MMP–2 IHC. (**c**) **Cholesteatoma**. Occasional MMP-9-positive cells in matrix and the perimatrix (arrows), MMP-9 IHC. (**d**) **Skin**. A moderate number of MMP-9-positive cells in the epithelium and connective tissue, MMP-9 IHC.

**Figure 5 medicina-59-00306-f005:**
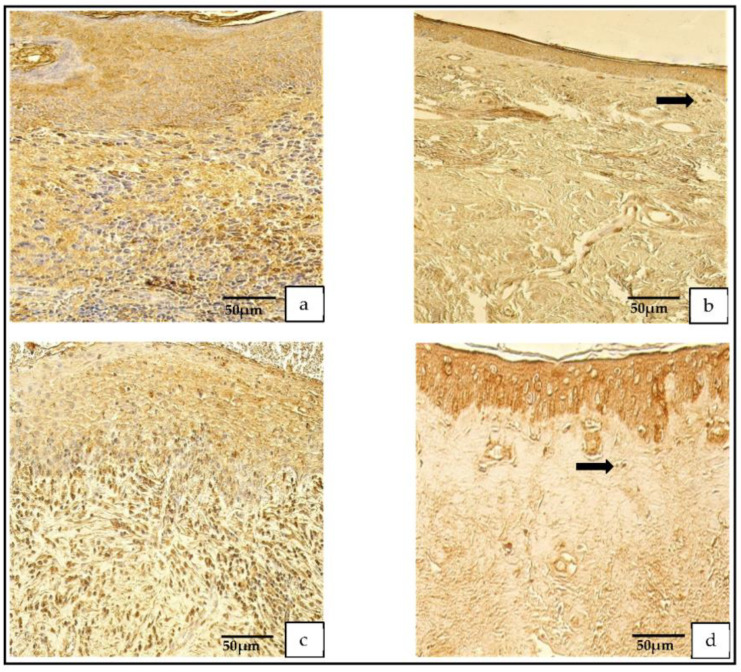
(**a**–**d**) Immunohistochemical micrographs. (**a**) **Cholesteatoma**. Numerous TIMP–2-positive cells in the matrix and moderate in the perimatrix, TIMP–2 IHC. (**b**) **Skin**. Moderate-to-numerous TIMP–2-positive cells in the epithelium and a few in the connective tissue (arrow), TIMP–2 IHC. (**c**) **Cholesteatoma.** Moderate-to-numerous TIMP-4-positive cells in matrix and numerous in the perimatrix, TIMP-4 IHC. (**d**) **Skin.** Moderate-to-numerous TIMP-4-positive cells in the epithelium and a few in the connective tissue (arrow), TIMP-4 IHC.

**Figure 6 medicina-59-00306-f006:**
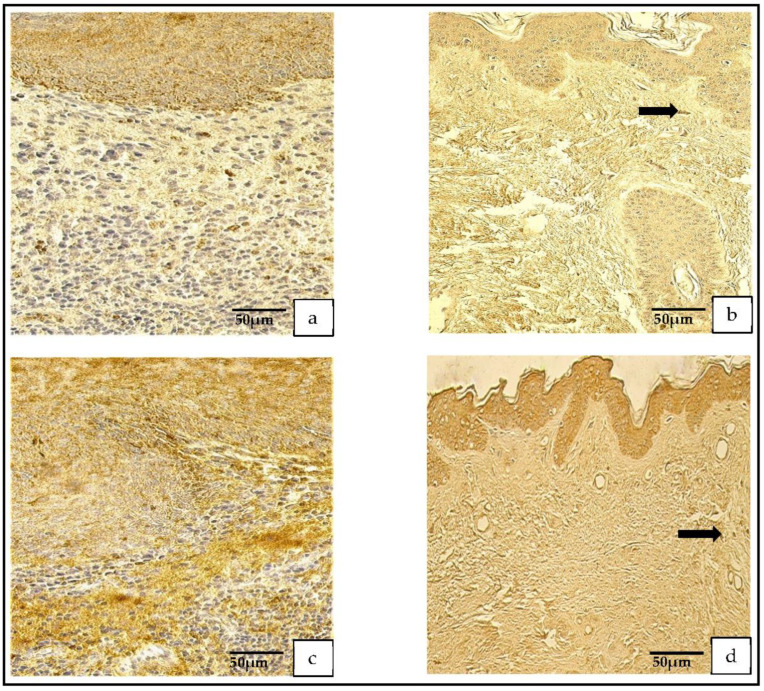
(**a**–**d**) Immunohistochemical micrographs. (**a**) **Cholesteatoma**. Numerous IL–1-positive cells in the matrix and in the perimatrix, IL–1 IHC. (**b**) **Skin.** Moderate-to-numerous IL–1-positive cells in the epithelium and a few in the connective tissue (arrow), IL–1 IHC. (**c**) **Cholesteatoma** Numerous IL–10-positive cells in the matrix and moderate in the perimatrix, IL–10 IHC. (**d**) **Skin.** Numerous IL–10-positive cells in the epithelium and a few in the connective tissue (arrow), IL–10 IHC.

**Figure 7 medicina-59-00306-f007:**
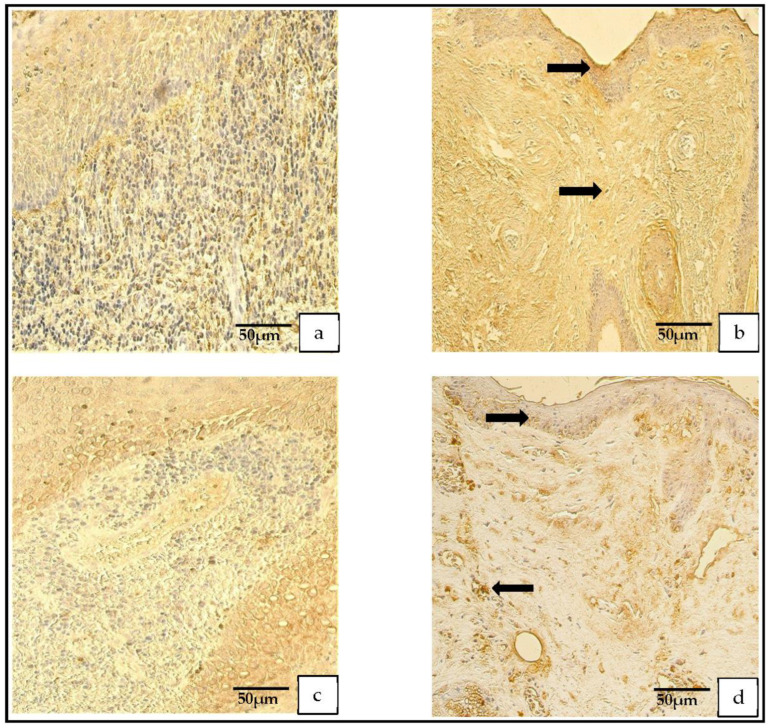
(**a**–**d**) Immunohistochemical micrographs. (**a**) **Cholesteatoma**. Moderate HβD–2-positive cells in matrix and perimatrix, HβD–2 IHC. (**b**) **Skin**. Few HβD–2-positive cells in the epithelium (arrow) and occasional HβD–2-positive cells in the connective tissue (arrow), HβD–2 IHC. (**c**) **Cholesteatoma.** Moderate-to-numerous HβD–4-positive cells in matrix and few-to-moderate in perimatrix, HβD–4 IHC. (**d**) **Skin**. A few HβD–4-positive cells in the epithelium (arrow) and occasional HβD–4-positive cells in the connective tissue (arrow), HβD–4 IHC.

**Figure 8 medicina-59-00306-f008:**
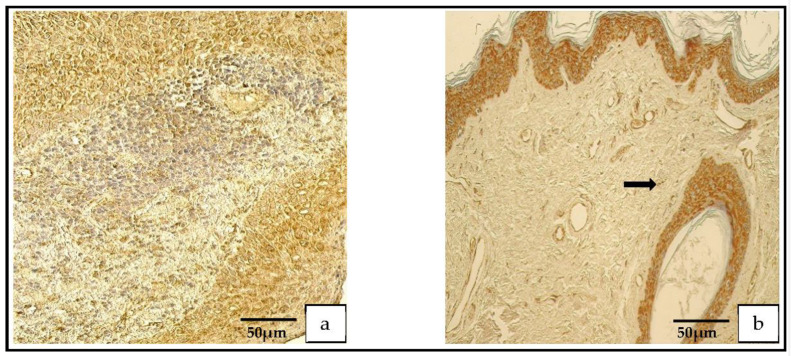
(**a**,**b**) Immunohistochemical micrographs. (**a**) **Cholesteatoma**. Numerous Shh-positive cells in the matrix and a moderate number in the perimatrix, Shh IHC. (**b**) **Skin**. Numerous-to-abundant Shh-positive cells in the epithelium and occasional Shh-positive cells in the connective tissue (arrow), Shh IHC.

**Table 1 medicina-59-00306-t001:** Relative numbers of different cell factors in the patient and control groups.

N	Age	Ki-67		NF-κβ		VEGF		MMP-2		MMP-9		TIMP-2		TIMP-4		IL-1		IL-10		HβD-2		HβD-4		Shh	
		M *	P *	M *	P	M	P	M	P	M *	P	M	P	M	P	M	P	M	P	M	P *	M	P	M	P *
P1	58	+	+	++	+++	++/+++	0/+	++/+++	+/++	0	+	+++	+	+/++	+++	+/++	+	++/+++	++	++/+++	++	++	+	+++	++
P2	46	0/+	00/+	+/++	+	+	0	+/++	+/++	0/+	0/+	0	0	+/++	+/++	+/++	+	++	+	++	++	0	++	++/+++	++
P3	23	0/+	00/+	++	++	+/++	0/+	0	+	0/+	0/+	0/+	+/++	++/+++	++/+++	++/+++	+++	+++	+++	++	+/++	+	0/+	++	++
P4	38	+	+	+++	+++	++/+++	0	+/++	0/+	0/+	0/+	0/+	0/+	++/+++	++/+++	+++	+/++	+++	++	++/+++	++	+/++	+	++/+++	++
P5	75	0/+	+	++	++	++	0	0/+	+/++	0	+	0	0/+	++/+++	++	++	+/++	++	++	++	+/++	0/+	0/+	++	+/++
P6	28	0/+	+/++	++	++	+++	++	0/+	+	+	+	00/+	00/+	+++	+++	++	++	+/++	++	+/++	+	0	00/+	+/++	++
P7	31	+	00/+	++/+++	++	++/+++	0/+	+	0/+	++	+	00/+	00/+	+++/++++	+++	0/+	0/+	+/++	+	+	+	00/+	00/+	++	++
P8	38	0/+	0/+	++	0/+	++/+++	+/++	+/++	+	0	0/+	+/++	0/+	++/+++	++	+/++	+	++/+++	+/++	++	+	0	00/+	+++	+
P9	26	+	+	+	++	+	+	+	+/++	0/+	+	0/+	0/+	++/+++	+++	+/++	++	+/++	+/++	+/++	+/++	0	00/+	++	++/+++
P10	39	0/+	0	+	0	+/++	0	00/+	00/+	0	0	0	0	++	+/++	0/+	00/+	0/+	0/+	+	0	0	0	+	0/+
P11	22	00/+	0/+	+++	++	++/+++	+/++	++	++	+	+/++	++	++	++/+++	++	++	+/++	++/+++	++	++/+++	+	0	00/+	++/+++	+/++
P12	19	00/+	0	++	+/++	++/+++	++	+/++	+/++	+	+	++	++	+++	++	++	+	++/+++	++	+/++	+	0	00/+	+++	+
P13	45	0/+	0	+	0	+/++	0	++	++	0	00/+	0	0	0	0	0	0	0/+	0/+	0/+	0/+	0	0	+/++	++
P14	24	+	0	+/++	0/+	++/+++	0/+	++	+/++	0	0	0	00/+	++/+++	+/++	0	0	+	0/+	+	+	+	+	+++	++/+++
P15	39	0/++	00/+	++/+++	0/+	0/++	00/+	++/+++	+/++	0	00/+	0/+	00/+	+++	++	0/+	+/++	0	0/+	+	00/+	++	0/+	+++	++
P16	40	00/+	0/+	++	00/+	00/+	0/+	++	+/++	0	0/+	+/++	00/+	+++	+/++	+/++	0/+	0/+	00/+	+/++	00/+	++	0/+	+++	+/++
P17	27	+/++	+/++	+++	0/+	+/++	+/++	+++/++++	+++	0	0/+	++/+++	0/+	+++/++++	++	0	+	0	+	+	++	+++	+/++	++++	+++
P18	41	0/+	00/+	++	00/+	0/+	00/+	++	+	0/+	+	+	00/+	+++	++	+/++	00/+	+	00/+	+/++	0/+	+/++	00/+	++/+++	++
P19	32	00/+	0/+	+/++	+	00/+	0/+	+/++	+	0/+	+/++	0	00/+	+++	++	0	+/++	0	0	+	0/+	0	00/+	+++	+/++
AVG	36.37	0/++	0/+	++	+/++	+/++	0/++	+/++	+/++	0/+	0/++	+	0/+	++/+++	++	+	+	+/++	+	+/++	+	0/++	0/+	++/+++	++
M–W	p	**0.000**	**0.010**	**0.001**	0.055	0.073	0.497	0.279	0.073	**0.008**	0.866	0.055	0.209	0.188	0.152	0.120	0.231	0.534	0.395	0.094	**0.004**	0.169	0.461	0.188	**0.000**
		**Ki-67**		**NF-κβ**		**VEGF**		**MMP-2**		**MMP-9**		**TIMP-2**		**TIMP-4**		**IL-1**		**IL-10**		**HβD-2**		**HβD-4**		**Shh**	
**N**	-	**E**	**CT**	**E**	**CT**	**E**	**CT**	**E**	**CT**	**E**	**CT**	**E**	**CT**	**E**	**CT**	**E**	**CT**	**E**	**CT**	**E**	**CT**	**E**	**CT**	**E**	**CT**
C1	-	0	0	0	00/+	++/+++	+	0/+	+	0/+	0/+	0/+	+	++	+	+	0/+	+	+	+	0/+	+	0/+	0	0
C2	-	0	0	0/+	0/+	++/+++	+	0	+	+/++	+	+	+	++/+++	++	0	+	++	++	+	0	+	0/+	++	+
C3	-	0	0	0	0	++	0/+	0	0/+	0/+	0/+	+	0	+/++	+/++	0/+	+	++	++	00/+	0	+/++	+	0/+	0
C4	-	00/+	00/+	+	0/+	+++	+/++	+++	+	+/++	0/++	++/+++	+	++/+++	++	0/+	0/+	++/+++	++	++	0/+	+	0/+	+++/++++	+
C5	-	0	0	++	+/++	+++	+	++/+++	+	+/++	+/++	++/+++	+	+++	++	0	0/+	++/+++	++	++	0/+	++	+	+++/++++	+
C6	-	00/+	00/+	+	0/+	++	0/+	+	+	+	0/+	++	0/+	++/+++	++	+	0/+	+	0/+	0/+	0	+	0/+	+/++	0/+
C7	-	0	0	0/+	0	++	0	0/+	+	+	0/+	+/++	0/+	+	+	+/++	+	+/++	+	0/+	00/+	+	0	0/+	+
AVG	-	0	0	0/++	0/+	++/+++	0/++	+	+	+	0/++	+/++	0/++	++	+/++	0/++	0/++	++	+/++	+	00/+	+	0/+	**+/++**	0/+

Abbreviations: P1–P19—patients 1–19; C1–C7—controls 1–7; AVG—average; M—matrix; P—perimatrix; E—epithelium; CT—connective tissue; Ki-67—proliferation marker; NF-κβ—nuclear factor kappa beta; VEGF—vascular endothelial growth factor; MMP-2—matrix metalloproteinase 2; MMP-9—matrix metalloproteinase 9; TIMP-2—tissue inhibitor of metalloproteinase-2; TIMP-4—tissue inhibitor of metalloproteinase-4; IL-1—interleukin 1; IL-10—interleukin 10; HβD-2—human beta defensin 2; HβD-4—human beta defensin 4; Shh—Sonic hedgehog gene protein. M–W—Mann–Whitney, p—p-value. 0 = no positive structures, 0/+ = occasional positive structures, + = few positive structures, +/++ = low-to-moderate number of positive structures, ++ = moderate number of positive structures, ++/+++ = moderate-to-numerous positive structures, +++ = numerous positive cells, +++/++++ = numerous-to-abundant structures, ++++ = abundance of positive structures in the visual field. *—statistically significant difference in the relative number in structures between subjects and controls.

**Table 2 medicina-59-00306-t002:** Statistically significant differences between patient and control groups.

Detected Factor	Mann–Whitney U Test	Z-Score	*p*-Value
Ki-67 matrix and Ki-67 control epithelium	5000	−3670	0.000
Ki-67 perimatrix and Ki-67 control connective tissue	23,000	−2604	0.010
NF-κβ matrix and NF-κβ control epithelium	13,000	−3176	0.001
MMP-9 matrix and MMP-9 control epithelium	22,000	−2679	0.008
HβD-2 perimatrix and HβD-2 control connective tissue	17,000	−2911	0.004
Shh perimatrix and Shh control connective tissue	8500	−3453	0.000

Abbreviations: Ki-67—proliferation marker; NF-κβ—nuclear factor kappa beta; MMP-9—matrix metalloproteinase 9; HβD-2—human beta defensin 2; Shh—Sonic hedgehog.

**Table 3 medicina-59-00306-t003:** Spearman’s rank correlations between different cell factors.

Markers		MMP-2 M	MMP-2 P	MMP-9 M	MMP-9 P	TIMP-2 M	TIMP-2 P	TIMP-4 M	TIMP-4 P	Shh M	Shh P	IL-1 M	IL-1 P	IL-10 M	IL-10 P	NF-κβ M	NF-κβ P	Ki-67 M	Ki-67 P	VEGF M	VEGF P	HβD2 M	HβD2 P	HβD4 M	HβD4 P
**MMP-2** **M**	Rs p																								
**MMP-2** **P**	Rs p	0.626 * 0.004																							
**MMP-9** **M**	Rs p	−0.383 0.105	−0.281 0.243																						
**MMP-9** **P**	Rs p	−0.119 0.627	0.003 0.989	0.635 * 0.004																					
**TIMP-2** **M**	Rs p	0.482 * 0.037	0.288 0.231	0.053 0.830	0.282 0.242																				
**TIMP-2** **P**	Rs p	0.024 0.922	0.205 0.399	0.233 0.338	0.466 * 0.044	0.697 * 0.001 *																			
**TIMP-4** **M**	Rs p	0.114 0.643	−0.111 0.652	0.334 0.162	0.309 0.198	0.283 0.241	0.108 0.660																		
**TIMP-4** **P**	Rs p	−0.267 0.270	−0.318 0.185	0.497 * 0.030	0.558 * 0.013	0.350 0.141	0.523 * 0.021	0.289 0.230																	
**Shh** **M**	Rs p	0.702 * 0.001	0.359 0.131	−0.275 0.255	0.017 0.946	0.543 * 0.016	0.309 0.199	0.358 0.132	−0.107 0.662																
**Shh** **P**	Rs p	0.312 0.194	0.291 0.227	−0.019 0.938	−0.162 0.507	0.006 0.981	−0.070 0.777	0.092 0.708	0.264 0.274	0.072 0.770															
**IL-1** **M**	Rs p	−0.425 0.070	−0.205 0.400	0.436 0.062	0.346 0.147	0.291 0.226	0.587 * 0.008	−0.102 0.676	0.438 0.061	−0.241 0.320	−0.249 0.304														
**IL-1** **P**	Rs p	−0.373 0.115	−0.063 0.796	0.367 0.122	0.439 0.060	0.128 0.600	0.513 * 0.025	0.115 0.640	0.631 * 0.004	−0.102 0.677	0.056 0.821	0.583 * 0.009													
**IL-10** **M**	Rs p	−0.354 0.138	−0.186 0.446	0.382 0.107	0.261 0.280	0.285 0.238	0.643 * 0.003	−0.352 0.140	0.431 0.065	−0.149 0.543	−0.164 0.503	0.820 * 0.000 *	0.366 0.123												
**IL-10** **P**	Rs p	−0.360 0.130	0.016 0.948	0.334 0.162	0.279 0.247	0.321 0.180	0.734 * 0.000	−0.206 0.397	0.580 * 0.009	−0.204 0.402	−0.009 0.971	0.777 * 0.000	0.600 * 0.007	0.839 * 0.000											
**NF-κβ** **M**	Rs p	0.311 0.195	0.055 0.823	0.232 0.339	0.232 0.340	0.592 * 0.008	0.507 * 0.027	0.518 * 0.023	0.392 0.097	0.359 0.132	0.072 0.769	0.350 0.142	0.269 0.265	0.249 0.305	0.358 0.132										
**NF-κβ** **P**	Rs p	−0.279 0.248	−0.123 0.616	0.499 * 0.030	0.563 * 0.012	0.210 0.388	0.624 * 0.004 *	−0.028 0.911	0.804 * 0.000	−0.091 0.711	0.163 0.506	0.627 * 0.004	0.674 * 0.002	0.690 * 0.001	0.779 * 0.000 *	0.389 0.100									
**Ki-67** **M**	Rs p	0.014 0.954	−0.068 0.783	−0.128 0.602	−0.185 0.449	0.015 0.952	0.030 0.903	−0.108 0.660	0.358 0.132	−0.073 0.766	0.683 * 0.001	−0.155 0.527	−0.129 0.599	0.120 0.625	0.192 0.431	0.040 0.870	0.276 0.252								
**Ki-67** **P**	Rs p	0.040 0.870	0.096 0.697	0.028 0.911	0.449 0.054	0.352 0.139	0.372 0.117	0.213 0.382	0.579 * 0.009	0.137 0.577	0.235 0.333	0.304 0.206	0.585 * 0.009	0.128 0.600	0.364 0.126	0.406 0.084	0.538 * 0.017	0.302 0.209							
**VEGF** **M**	Rs p	−0.171 0.484	−0.105 0.668	0.274 0.256	0.110 0.654	0.162 0.507	0.399 0.091	−0.101 0.681	0.386 0.103	−0.132 0.590	−0.028 0.908	0.361 0.128	0.071 0.773	0.588 * 0.008	0.674 * 0.002	0.315 0.189	0.512 * 0.025	0.328 0.171	0.145 0.553						
**VEGF** **P**	Rs p	0.070 0.776	0.192 0.431	0.368 0.121	0.428 0.068	0.581 * 0.009	0.549 * 0.015	0.475 * 0.040	0.403 0.087	0.346 0.146	0.012 0.963	0.128 0.601	0.293 0.224	0.139 0.570	0.308 0.200	0.265 0.273	0.221 0.364	−0.059 0.811	0.308 0.199	0.384 0.105					
**HβD** **-** **2** **M**	Rs p	−0.128 0.602	−0.048 0.845	0.205 0.399	0.372 0.117	0.416 0.077	0.616 * 0.005	−0.322 0.178	0.376 0.112	0.040 0.870	−0.186 0.446	0.822 * 0.000	0.463 * 0.046 *	0.841 * 0.000	0.694 * 0.001 *	0.346 0.147	0.654 * 0.002	0.000 0.998	0.431 0.066	0.326 0.173	0.078 0.752				
**HβD** **-** **2** **P**	Rs p	−0.033 0.893	0.189 0.438	0.140 0.566	0.197 0.419	0.236 0.331	0.491 * 0.033	−0.201 0.410	0.455 0.051	0.118 0.631	0.456 * 0.050	0.382 0.107	0.382 0.106	0.583 * 0.009	0.663 * 0.002 *	0.268 0.267	0.692 * 0.001	0.618 * 0.005	0.520 * 0.022	0.371 0.118	0.100 0.683	0.577 * 0.010			
**HβD** **-** **4** **M**	Rs p	0.538 * 0.018	0.143 0.560	−0.414 0.078	−0.232 0.340	0.412 0.080	0.145 0.554	0.313 0.192	0.110 0.653	0.483 * 0.036	0.420 0.073	−0.065 0.792	−0.096 0.696	−0.160 0.513	−0.094 0.703	0.520 * 0.022	0.017 0.945	0.297 0.218	0.247 0.307	−0.186 0.447	−0.149 0.544	0.040 0.870	0.165 0.500		
**HβD** **-** **4** **P**	Rs p	0.350 0.142	0.269 0.266	−0.242 0.317	−0.185 0.448	0.184 0.450	0.182 0.456	0.000 0.998	0.048 0.846	0.525 * 0.021	0.474 * 0.040	0.107 0.664	0.138 0.573	0.161 0.511	0.184 0.452	0.355 0.136	0.302 0.209	0.356 0.134	0.328 0.171	−0.011 0.963	−0.118 0.631	0.353 0.138	0.650 * 0.003	0.640 * 0.003	

Abbreviations: Rs—Spearman’s correlation coefficient; p—p-value; M—matrix; P—perimatrix; MMP-2—matrix metalloproteinase 2; MMP-9—matrix metalloproteinase 9; TIMP-2—tissue inhibitor of metalloproteinase-2; TIMP-4—tissue inhibitor of metalloproteinase-4; Shh—Sonic hedgehog gene protein; IL-1—interleukin 1; IL-10—interleukin 10; NF-κβ—nuclear factor kappa beta; Ki-67—proliferation marker; VEGF—vascular endothelial growth factor; HβD-2—human beta defensin 2; HβD-4—human beta defensin 4. * Correlation is significant at the 0.05 level (two-tailed).

## Data Availability

The datasets used and/or analyzed during the current study are presented in the results section.

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
