# Peer review of "Morphopathogenesis of Adult Acquired Cholesteatoma"

_medicina, 2023, doi:10.3390/medicina59020306_

Round 1

Reviewer 1 Report

In this manuscript the authors compared the distribution of proliferation markers (Ki-67, NF-κβ), tissue remodeling factors (MMP-2, MMP-9, TIMP-2, TIMP-4), vascular endothelial growth factor (VEGF), Interleukins (IL-1 and IL-10), human beta defensins (HβD-2 and HβD-4) and Sonic hedgehog gene protein in cholesteatoma and control skin. They performed statistical analyzes and found some statistically perceptible differences between some measured parameters.

The major weakness of this work, of which the authors are aware, is the small number of samples used for the study (19 from patients with cholesteatoma and 7 control skin samples). Nevertheless, the authors analyzed many microscopic samples. After reading their manuscript I have some specific comments.

1.       Some more information should be added in the introduction, I mean especially the one about Ki-67 and NF-κß.

2.       Very inadequately described methodology: section 2.2. Immunohistochemical Analysis. The sentence “The Biotin-Streptavidin biochemical method was used for immunohistochemistry to detect:…” is too vague. This Biotin-Streptavidin method should be described in more detail: what secondary antibodies were used, what were they conjugated to (I assume that it was HRP) and what kind of substrate was used (I suppose that it was one that produces a signal in the visible light range). The name and manufacturer of these compounds should be provided, as well some technical details.

3.       All Figures 1-8 (Histological micrographs and  Immunohistochemical micrographs) are presented without details of data during colleting photos, i.e. magnification (this can be presented as a scale in the image or as a lens magnification value). These values are required for proper comparison of test samples and controls.

4.       An additional negative control should be provided to show that the proper signals were analyzed: blocking buffer alone (without antibody) with the test sample of the cholesteatoma as a negative control. This is evidence that there are no false positive signals.

5.       The conclusions lack an overall summary of the studies: their significance and future perspectives.

I proposed a major revision for the manuscript for because I assumed that the authors only make mistake in the presentation of the methodology (section 2.2. Immunohistochemical Analysis). 

Reviewer 2 Report

The paper provides very interesting data but it still needs a considerable revision to be acceptable for the Medicina.

Line 127:

Incorporating labelling positive structures information into the Figure legend facilitates comprehension. A picture of the tissue at each criterion would be helpful for the reader to understand.

Line 140:

Is this statistical methodology comparable to that found in other literature? It would be prudent to examine the validity of this ordinal variable.

Table 1:

Given that the statistical methodology employed is the Mann Whitney method, it is imperative to specify the median and interquartile range.

Result:

A middle ear cholesteatoma comprises three layers: the perimatrix, the matrix, and cystic contents (Kitaya S, et al. Laryngoscope Investig Otolaryngol. 2022 Jul 8;7(4):1155-1163). The cystic content should also be subject to examination.

Round 2

Reviewer 1 Report

Below, I showed how the authors responded to my remarks. Some points still demand improvement

Ad. 1. OK

Ad. 2. ok

Ad. 3. Methodology. This part has been changed and now it is understandable. However, I still cannot find the manufacturer of HiDef Detection™ HRP Polymer System. Was it Cell Marque, USA? Also, the authors give the LOT number, whcich is inappropriate and unnecessary. Instead of LOT, a  catalog number can/should be given. 

Additionally it is written that “HRP chromogene”(here is also spelling error, it should be chromogen) was used, but no one knows which one. The chromogen is not supplied together with HiDef Detection Kit and should be listed separately as there are different substrates for HRP.

Ad.4. Adding  a scale bar is very useful and helpful.  However, the authors wrote: “Also, we left the lens magnification value as it was originally written at the Figure legends”. Now, I have noticed the lens magnification in the Figure legends. But these magnifications are given in the unacceptable form “X 200” – this is colloquialism and should be changed to  scientific language.

Ad. 5. Ok

Ad.6.  There is still lack of overall summary of the studies in the conclusions. Instead, the authors added a summary at the end of the Discussion, which is inappropriate.

Reviewer 2 Report

The manuscript has been much improved and is in a nice condition now.
